# Comparison of Feeding Efficiency and Hospital Mortality between Small Bowel and Nasogastric Tube Feeding in Critically Ill Patients at High Nutritional Risk

**DOI:** 10.3390/nu12072009

**Published:** 2020-07-06

**Authors:** Wei-Ning Wang, Chen-Yu Wang, Chiann-Yi Hsu, Pin-Kuei Fu

**Affiliations:** 1Department of Food and Nutrition, Taichung Veterans General Hospital, Taichung 40705, Taiwan; sherry@vghtc.gov.tw; 2Department of Critical Care Medicine, Taichung Veterans General Hospital, Taichung 40705, Taiwan; chestmen@gmail.com; 3Department of Nursing, Hungkuang University, Taichung 43302, Taiwan; 4Biostatistics Task Force of Taichung Veterans General Hospital, Taichung 40705, Taiwan; chiann@vghtc.gov.tw; 5College of Human Science and Social Innovation, Hungkuang University, Taichung 43302, Taiwan; 6Department of Computer Science, Tunghai University, Taichung 40705, Taiwan

**Keywords:** critically ill patients, feeding efficiency, high nutritional risk, hospital mortality, small bowel enteral nutrition

## Abstract

Nasogastric tube enteral nutrition (NGEN) should be initiated within 48 h for patients at high nutritional risk. However, whether small bowel enteral nutrition (SBEN) should be routinely used instead of NGEN to improve hospital mortality remains unclear. We retrospectively analyzed 113 critically ill patients with modified Nutrition Risk in Critically Ill (mNUTRIC) score ≥ 5 and feeding volume < 750 mL/day in the first week of their stay in the intensive care unit (ICU). Age, sex, mNUTRIC score, and Acute Physiology and Chronic Health Evaluation II (APACHE II) score were matched in the SBEN (*n* = 48) and NGEN (*n* = 65) groups. Through a univariate analysis, factors associated with hospital mortality were SBEN group (hazard ratio (HR), 0.56; 95% confidence interval (CI), 0.31–1.00), Simplified Organ Failure Assessment (SOFA) score on day 7 (HR, 1.12; 95% CI, 1.03–1.22), and energy intake achievement rate < 65% (HR, 2.53; 95% CI, 1.25–5.11). A multivariate analysis indicated that energy intake achievement rate < 65% on the third follow-up day (HR, 2.29; 95% CI, 1.12–4.69) was the only factor independently associated with mortality. We suggest initiation of SBEN on the seventh ICU day before parenteral nutrition initiation for critically ill patients at high nutrition risk.

## 1. Introduction

Malnutrition and undernutrition have been recognized as a highly prevalent problem during intensive care unit (ICU) stays [1,2]. Up to 50–60% of critically ill patients are reported to experience energy and protein deficits during their ICU stay, which are correlated with morbidity and mortality [3,4]. No consensus exists regarding a gold standard tool for identifying ICU patients who are at high risk and are malnourished [5]. However, screening tools such as the Nutritional Risk Score, Nutrition Risk in Critically Ill (NUTRIC) assessment score, and modified NUTRIC (mNUTRIC) score have been widely applied to identify high-risk patients and enable initiation of early enteral nutrition (EN) therapy in the ICU [6,7,8,9,10,11,12]. Patients with mNUTRIC scores of more than 5 points are identified as being at high nutritional risk and urgently requiring additional energy support to reduce the likelihood of mortality [8,13,14]. In our previous study, an energy achievement rate of lower than 65% in patients admitted to ICU was associated with a higher risk of mortality [15], and targeted energy intake was also a mortality predictor in this population [13]. Therefore, a systemic approach to recognizing ICU patients at high nutritional risk and initiating early EN in these patients has become the standard of care for decreasing morbidity, pneumonia incidence, and even hospital mortality in critically ill patients [5,6,7,16,17,18].

The current guidelines recommend that early nasogastric tube enteral nutrition (NGEN) should be initiated within 48 h of ICU admission because this route is easier to facilitate initiation of EN as soon as possible [5,6,7,11]. However, small bowel EN (SBEN) has theoretical advantages over stomach EN because it not only decreases the risks of gastroesophageal reflux and aspiration pneumonia, but also delivers more nutrients than does NGEN [19,20,21,22,23]. The current guidelines including the European Society for Clinical Nutrition and Metabolism (ESPEN), the American Society for Parenteral and Enteral Nutrition (ASPEN), and the Society of Critical Care Medicine (SCCM) guidelines strongly recommend NGEN for EN initiation in critically ill patients at high risk and a shift to SBEN when the patient has developed intolerance of NGEN despite the administration of prokinetic agents [5,6,7]. For patients unable to meet the >60% achievement rate for protein and energy intake through the EN alone, supplemental parenteral nutrition (PN) can be considered after 7–10 days of EN [6,7].

Our previous study suggested the evaluation of the role of SBEN on the seventh day of the ICU stay for those with energy achievement rates of less than 60% [9]. We demonstrated that the feeding volume and achievement rates (%) of protein and energy intake were significantly higher after SBEN on the 10th day of the ICU stay [8]. We also found that an energy achievement rate lower than 65% on the third day after SBEN (the 10th day of the ICU stay) increased the mortality risk by a factor of 4.97 [8]. However, whether SBEN should be routinely used instead of NGEN for undernourished critically ill patients and to improve the hospital mortality remains unclear. The current retrospective study compared the feeding efficiency and hospital mortality of two feeding routes (maintaining NGEN or shifting to SBEN) on the seventh ICU day in ICU patients who were undernourished and at high nutritional risk.

## 2. Materials and Methods

### 2.1. Study Design and Ethic Approval

This retrospective, case–control study was conducted in the respiratory intensive care unit (RICU) of Taichung Veterans General Hospital from January 2014 to December 2015 in Central Taiwan. The Institutional Review Board of Taichung Veterans General Hospital reviewed and approved the study protocol (IRB number, CE16028A; date of approval, 29 January 2016). The current study waived the requirement for informed patient consent due to the availability of all delinked data from an electronic medical record system and the study design was a retrospective analysis.

### 2.2. Patient, Clinical Setting, Nutrition Evaluation, and Routine

The RICU is an adult medical ICU with 24 beds servicing patients with pneumonia accompanied by various conditions—such as septic shock and acute respiratory distress syndrome—of various severities. It has a full-time registered dietitian to conduct nutritional risk and target evaluation and provide suggestions for routine application. Serial information including mNUTRIC score, nutritional goals, and the achievement rate of energy target are routinely determined for each critically ill patient by the full-time dietitian. The mNUTRIC score has five components: age, number of comorbidities, Simplified Organ Failure Assessment (SOFA) score, Acute Physiology and Chronic Health Evaluation II (APACHE II) score, and days in hospital before ICU admission [24]. For every patient admitted to the RICU, the dietitian determines their mNUTRIC score (0–9 points) within 48 h of ICU admission and records it in the electronic medical record, as described in our previous study [8,13,15]. The target energy requirement was 25–30 kcal/kg/day and the protein intake target was 1.2 g/kg/day in accordance with the 2016 ASPEN/SCCM guidelines [6,7].The achievement rate (%) of energy was defined as follows: (actual energy intake/estimated energy requirement) × 100; the achievement rate (%) of protein was adjusted as follows: (actual protein intake/estimated protein requirement) × 100 [8,13,15].

Early EN was initiated on the first day for all patients admitted to the RICU except in some clinical contraindications. On the seventh ICU day, a transpyloric tube for SBEN was an option in patients with high nutritional risk (mNUTRIC ≥ 5) and severe malnutrition (energy target achievement rate < 30% or feeding volume < 750 mL/day) after trying prokinetic agents. Some of these patients were shifted to SBEN, but NGEN feeding was retained for others, because the use of SBEN for undernourished patients on the seventh ICU day is not mandatory according to the current guidelines. These clinical situations thus enabled us to analyze the two feeding routes on the seventh ICU day and their effects on feeding efficiency and mortality. From January 2014 to December 2015, a total of 113 patients with malnutrition and at high nutritional risk were retrospectively enrolled in the analysis and were classified into SBEN (*n* = 48) and NGEN (*n* = 65) groups through matching according to age, sex, mNUTRIC score, and APACHE II score (Appendix A revealed the study flow.).

### 2.3. Data Collection, Adjustment, and Outcome Measures

Data were collected on age, sex, body mass index (BMI), severity of illness (SOFA and APACHE II scores), mNUTRIC score, major comorbidities, and Charlson Comorbidity Index (CCI). The primary outcome was feeding efficiency, including average energy intake achievement rate (%); hospital mortality was the secondary outcome. The index date was the seventh day of the ICU stay, which was the day defining whether a patient with critical illness and malnutrition was placed in the SBEN or NGEN group. Those who had received transpyloric tube insertion were enrolled into the SBEN group. Daily energy intake was recorded from the index day (the seventh ICU day) until 7 days later (the 14th ICU day). As in our previous report [8], the serial nutrition parameters 3 days after the index date were employed to compare the feeding efficiency of the SBEN and NGEN groups. Feeding efficiency was evaluated by the following six parameters: (1) actual feeding volume (mL/day), (2) actual energy intake (kcal/day), (3) actual protein intake (g/day), (4) actual protein intake (g/kg/body weight (BW)), (5) energy intake achievement rate (%), and (6) protein intake achievement rate (%). The primary outcome was the comparison of feeding efficiency between the SBEN and NGEN groups. The secondary outcome was factors associated with hospital mortality in ICU patients with mNUTRIC score ≥ 5 and feeding volume < 750 mL/day. We also validated the power of hospital mortality prediction in this setting because an energy achievement rate of < 65% on the third day after the index date (the 10th ICU day) was demonstrated to be an outcome predictor in our other study [8].

### 2.4. Statistical Analysis

Categorical variables are presented as frequencies and percentages. The chi-squared test was used to determine significance. The Mann–Whitney U test was applied to determine differences between groups for data with nonparametric distribution, and the results are presented as the median and interquartile range (IQR). Factors associated with hospital mortality were assessed through Cox regression analysis. The strength of an association is presented as the hazard ratio (HR) and 95% confidence interval (CI). All tests were two sided, with *p* < 0.05 considered significant. The SPSS statistical software package (version 24.0; International Business Machines Corp, Armonk, NY, USA) was used for all statistical analyses.

## 3. Results

Table 1 reveals a comparison of demographic characteristics, severity scores, comorbidities, energy intake, and hospital mortality between the SBEN and NGEN groups. The median mNUTRIC score of the cohort was 7 (IQR, 5.25–8 in the SBEN group and 5.5–8 in the NGEN group), indicating that these patients were at high nutritional risk. The APACHE II (27.0 and 30.0 for the SBEN and NGEN groups, respectively) and SOFA (10 and 10 for the SBEN and NGEN groups, respectively) scores also indicated the high clinical severity of disease in this cohort. However, cardiovascular disease (64.58% and 26.15% in the SBEN and NGEN groups, respectively, *p* < 0.001) and cancer (45.83% and 12.31% in the SBEN and NGEN groups, respectively, *p* < 0.001) were more common in the SBEN group than the NGEN group; liver cirrhosis (22.92% and 55.38% in the SBEN and NGEN groups, respectively, *p* = 0.001) was less common in the SBEN group. The overall hospital mortality rate exhibited no significant difference between these two groups (43.75% and 43.08% in the SBEN and NGEN groups, respectively, *p* > 0.05). The achievement rate (%) of daily average energy intake on days 2–7 was significantly higher in the SBEN group than the NGEN group on each observation day. As shown in Table 1, the average energy achievement rate was higher than 60% in the SBEN group on the fourth day, but remained lower than 60% until the seventh day in the NGEN group. Using the energy achievement index parameter of more than 65% on the third day (the 10th ICU day) as the objective goal, 45.83% of patients achieved the goal in the SBEN group, but only 26.15% reached the goal in the NGEN group, a significant difference (*p* = 0.048).

Table 2 presents a comparison of the feeding efficiency of the SBEN and NGEN groups. Actual daily energy parameters represent the feeding efficiency on the third observation day, including actual feeding volume (mL/day), actual energy intake (kcal/day), actual protein intake (g/day), actual protein intake (g/kg/BW), energy intake achievement rate (%), and protein achievement rate (%). All feeding parameters were significantly higher in the SBEN group than the NGEN group (Table 2). The energy intake achievement rate (%) in the SBEN group significantly increased day by day compared with that in the NGEN group (*p* = 0.021; Figure 1).

The parameters of surviving and nonsurviving patients were compared (Table 3). The nonsurviving patients had higher SOFA scores on the seventh ICU day and lower energy intake achievement rates (%) on each day subsequent to the enrollment day. Age, sex, BMI, mNUTRIC score, APACHEII score, and comorbidities were not significantly different between the survival and nonsurvival groups. In the survival group, the median energy intake was 60% on the fourth day (the 11th ICU day). By contrast, the median energy intake achievement in the nonsurvival group remained at lower than 50%, even on the seventh day after enrollment (the 14th ICU day; Table 3).

Factors associated with mortality by Cox regression analysis are shown in Table 4. A univariate analysis identified only three factors associated with mortality: SBNE group (HR, 0.56; 95% CI, 0.31–0.997; *p* = 0.049), SOFA score on the seventh ICU day (HR, 1.12; 95% CI, 1.03–1.22; *p* = 0.009), and energy achievement rate < 65% on the third day after enrollment (the 10th ICU day; HR, 2.53; 95% CI, 1.25–5.11; *p* = 0.01). A multivariate analysis indicated that energy intake achievement rate < 65% on the third day after enrollment (HR, 2.29; 95% CI, 1.12–4.69) was the only predictive factor that differed significantly between the survival and nonsurvival groups. In this cohort, an energy achievement rate > 65% was associated with lower 60-day, 90-day, and hospital mortality after adjustment for age, sex, BMI, and SOFA and APACHE II scores in patients at high nutritional risk (mNUTRIC ≥ 5; Figure 2).

## 4. Discussion

The present study reports three major findings regarding critically ill patients admitted to the ICU with malnourishment and high nutritional risk. First, after 7 days of stomach EN, the initiation of adjuvant SBEN improved energy delivery and had potential benefits in relation to hospital mortality compared with continued stomach EN. Second, the feeding efficiency increased significantly after SBEN initiation and resulted in the nutritional goal being achieved within 3 days. Third, achievement of 65% of the energy requirements by day 3 after SBEN initiation (on the 10th ICU day) was the key factor associated with survival. If a patient does not reach the goal within 3 days of SBEN initiation, we suggest that PN should be considered to improve the energy intake achievement rate.

The current guidelines—including the ASPEN/SCCM and ESPEN guidelines and the consensus statement of the Asia-Pacific and Middle East regions—recommend that every critically ill patient staying in the ICU for more than 48 h should be considered as being at risk of malnutrition, and a systemic approach to, as well as evaluation of, nutritional risk is suggested [5,6,11]. EN should be initiated within 48 h to provide nutritional support to critically ill patients who are unable to maintain volitional intake [5,6,7]. The current evidence does not suggest routine feeding through SBEN [19]. Accordingly, NGEN is recommended as the standard approach for early EN, and SBEN can be used as adjuvant therapy for those with gastric feeding intolerance that cannot be resolved with prokinetic agents [5,6,7]. In fact, studies have demonstrated that SBEN provides greater feeding efficiency than NGEN [19,20,21,23]. However, the right time to shift from NGEN to SBEN remains unclear in the current guidelines. Regarding actual clinical practice, we conduct a feeding protocol to evaluate the energy intake achievement rate at the end of the first week of an ICU stay, and this evaluation enables initiation of SBEN on the eighth ICU day as an option in ICU care [8]. Therefore, all patients enrolled in the present cohort had high mNUTRIC score (median mNUTRIC score, 7; IQR, 5.25–8) and low energy intake achievement rate (<50% of the target goal or feeding volume < 750 mL/day) even though NGEN was started within 48 h of ICU admission. To the best of our knowledge, the present study, which initiated SBEN as adjuvant therapy on the 7th ICU day to improve energy delivery compared with maintenance of NGEN, is the first data-based report to examine critically ill, ICU patients at high nutritional risk.

Another major issue overlooked in the guidelines is the optimal time at which to decide that the SBEN approach has failed and feeding route should be switched from SBEN to supplemental PN or total PN. Our previous study suggested that on the third day after SBEN initiation, the optimal energy intake achievement rate was higher than 65% [8]. For critically ill patients for whom NGEN has failed and have been shifted to SBEN on the eighth ICU day, an energy achievement rate of <65% 3 days after SBEN initiation was significantly associated with increased mortality (odds ratio, 4.97; 95% CI, 1.44–17.07) [8]. The 2019 ESPEN guidelines recommend that PN should not be started until all strategies to maximize EN tolerance have been attempted [5]. Heidegger et al. suggested that starting supplementary PN on the fourth day after ICU admission may reduce nosocomial infections and improve clinical outcomes in patients for whom EN is insufficient [25]. Some studies have suggested initiating EN plus PN between the fourth and seventh days in the ICU [26,27,28]. However, the 2016 ASPEN/SCCM guidelines recommend considering the initiation of supplemental PN only if >60% energy and protein intake achievement are not achieved after 7–10 days of EN because initiating PN before this time has no benefits [6,7]. Braunschweig et al. published research on the timing and doses of energy received by patients with acute lung injury, inferring that higher later (days ≥ 8) energy intake may reduce the mortality risk [29]. Therefore, the optimal time point for administration of supplementary PN with the aim of meeting full energy requirements and whether SBEN should be attempted before a patient is shifted to PN remain unclear. Our data suggest that for critically ill patients with mNUTRIC score ≥ 5 and poor feeding achievement rate (<50% of target goal or feeding volume < 750 mL/day) using gastric tube feeding, SBEN for 3 days is an option prior to PN initiation.

The final point that merits discussion is whether SBEN has advantages in terms of mortality. The literature shows that adequate nutritional support has a significant mortality benefit in those at high risk of malnutrition and who are critically ill [14,30]. Our study conducted in 2019 also suggested that energy requirements achieving <65% within 3 days of SBEN initiation is the independent factor associated with mortality (OR, 4.97; 95% CI, 1.44–17.07) in malnourished and high nutritional risk populations [8]. Our previous studies suggested that critically ill patients at high nutritional risk who receive at least 65% of their energy intake requirements and consume at least 800 kcal/day exhibit significantly lower 14-day, 28-day, and hospital mortality rates [13,15]. However, in populations at low nutritional risk (mNUTRIC score < 5), copious evidence indicates no survival benefits from full nutritional support [13,24,30,31,32]. This phenomenon is due to the strong correlation of suboptimal energy delivery with increased nosocomial infection, longer duration of ventilator dependency, and longer ICU and hospital stays—these factors contribute to greater mortality [14,15,33]. The current study enrolled patients from ICUs who were elderly, had high disease severity, and were at high nutrition risk. Most were admitted to the RICU because of pneumonia-related sepsis, septic shock, acute respiratory failure, or acute respiratory distress syndrome. Therefore, adequate nutritional support played a major role in their survival. The nutritional intake cut-off value for distinguishing survival from nonsurvival was 65%, the same as that indicated by our other research [8,13]. The univariate analysis revealed that the SBEN group experienced mortality benefits, but had a higher SOFA score on the seventh ICU day, and energy intake achievement rate < 65% on the 10th ICU day was significantly associated with hospital mortality. However, in the multivariable analysis, the only factor that was discovered to independently predict hospital mortality was energy intake achievement rate < 65% on the 10th ICU day (HR, 2.29; 95% CI, 1.12–4.69). We suggest that the potential mortality benefits from SBEN initiation are due to energy intake improvements in patients with very high disease severity. As such, if energy intake achievement is not improved substantially 3 days after SBEN initiation, supplementary PN or total PN should be considered to increase nutrition support.

This study has several limitations. First, the study design was a retrospective case-controlled study, in which two groups from a database were matched, rather than a prospective randomized controlled trial. Thus, heterogeneity may have existed between the two groups. Second, the study was conducted at a single ICU in a medical center, meaning that the results may not be generalizable. In practice, feeding protocols and patient populations may vary between ICUs. In our RICU, we followed a feeding protocol approved by the registered dietitian, who also provided nutrition risk evaluation and nutrition therapy recommendations in daily practice. For patients with an energy target achievement rate < 50% or a feeding volume < 750 mL/day, our protocol has provided the option since 2015 of consulting a gastroenterologist to evaluate the suitability of transpyloric tube insertion for SBEN on the seventh ICU day. Therefore, heterogeneity in ICU care and feeding protocols was minimal in the present study. Third, critically ill patients in ICU may require image studies, such as computed tomography and magnetic resonance imaging, invasive procedures, and gastrointestinal endoscopy, which can interrupt feeding protocols. Finally, the present results may not be generalizable to critically ill patients in pediatric, cardiac, neurosurgical, and surgical ICUs or even general ICUs because the current study was conducted only in RICU patients.

## 5. Conclusions

Replacing NGEN with SBEN on the seventh ICU day significantly improves energy delivery and may reduce hospital mortality risk in patients at high nutritional risk. The key factor determining survival in this population is an energy intake requirement rate > 65% within 3 days of SBEN initiation. For malnourished, critically ill patients at high nutritional risk (mNUTRIC score ≥ 5) who have been treated with all other available strategies to maximize EN tolerance, we suggest initiating SBEN on the seventh ICU day before PN initiation, and the clinician should follow up on the energy intake achievement rate (at least >65%) 3 days after SBEN initiation to determine the subsequent step of nutrition support therapy.

## Figures and Tables

**Figure 1 nutrients-12-02009-f001:**
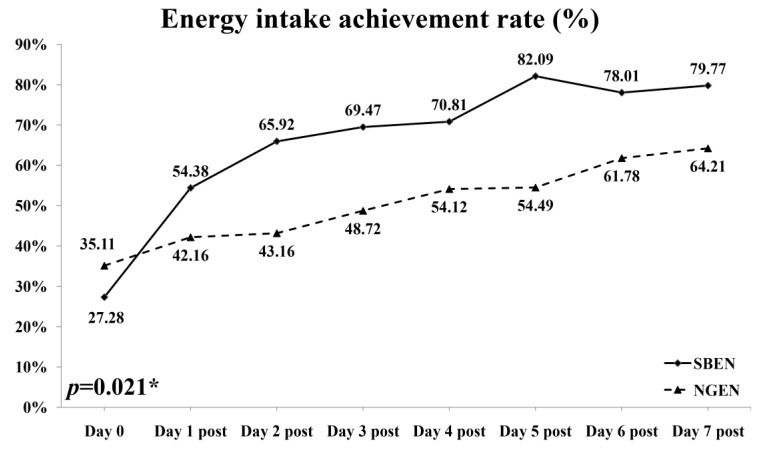
Achievement rate of energy intake for SBEN and NGEN groups. *p* = 0.021

**Figure 2 nutrients-12-02009-f002:**
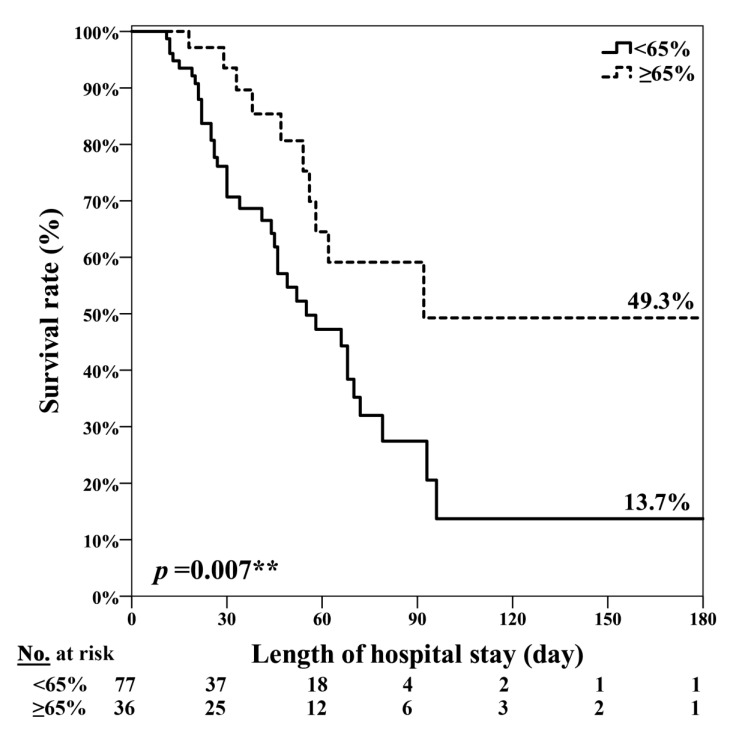
Energy achievement rate ≥ 65% on the 10th ICU day was significantly associated with lower hospital mortality in patients at high nutritional risk. ** *p* = 0.007.

**Table 1 nutrients-12-02009-t001:** Demographic characteristics, severity scores, comorbidities, and clinical outcomes for all malnourished patients in the intensive care unit (ICU).

Variables	SBEN Group (*n* = 48)	NGEN Group (*n* = 65)	*p*-Value
Age (year)	70.45 (59.75–79.94)	74.17 (54.63–85.12)	0.423
Gender—Male (*n*, %) ^a^	31 (64.58%)	48 (73.85%)	0.393
Body mass index (kg/m^2^)	23.62 (21.29–27.32)	22.77 (21.1–25.05)	0.238
mNUTRIC score	7.00 (5.25–8)	7.0 (5.5–8)	0.491
APACHE II score	27.00 (24–32.75)	30.00 (27–33)	0.156
SOFA-day 1	10.00 (8–12.75)	10.00 (6.5–13)	0.875
SOFA—day 3 (*n* = 47 vs. 64)	9.00 (6–12)	10.00 (8–13)	0.172
SOFA—day 7 (*n* = 46 vs. 64)	8.5 (5–11)	9.0 (7–11)	0.490
CCI	2.5 (1–4)	0.0 (0–2)	<0.001 **
Comorbidity (*n*, %) ^a^			
Diabetes mellitus	13 (27.08%)	20 (30.77%)	0.828
Chronic kidney disease	9 (18.75%)	14 (21.54%)	0.898
Cardiovascular disease	31 (64.58%)	17 (26.15%)	<0.001 **
Cancer	22 (45.83%)	8 (12.31%)	<0.001 **
Liver cirrhosis	11 (22.92%)	36 (55.38%)	0.001 **
Average energy intake achievement rate (%) after enrollment			1.000
2nd day	57.16 (36.29–70.69)	41.35 (30.16–65.12)	
3rd day	55.91 (41.5–75.49)	44.16 (30.85–64.41)	0.039 *
4th day	60.29 (41.29–75.11)	49.02 (32.36–66.74)	0.015 *
5th day	63.36 (43.2–78.62)	49.45 (32.05–69.46)	0.026 *
6th day	63.63 (45.5–81.01)	51.12 (32.24–71.99)	0.015 *
7th day	65.20 (48.91–84.47)	55.12 (32.57–75.35)	0.022 *
Average energy intake achievement rate (%) ≥ 65% (3 days after the index day) (*n*, %) ^a^	22 (45.83%)	17 (26.15%)	0.031 *
Hospital mortality (*n*, %) ^a^	21 (43.75%)	28 (43.08%)	0.048 *

^a^ Chi-squared test or Mann–Whitney U test, * *p* < 0.05, ** *p* < 0.01. SBEN, small bowel enteral nutrition; NGEN, nasogastric tube enteral nutrition; mNUTRIC, modified Nutrition Risk in Critically Ill; APACHE II, Acute Physiology and Chronic Health Evaluation II; SOFA, Sequential Organ Failure Assessment; CCI, Charlson Comorbidity Index. Categorical variables are presented as frequencies and percentages (%).

**Table 2 nutrients-12-02009-t002:** Comparison of feeding efficiency between SBEN and NGEN groups on the third day after enrollment in malnourished, critically ill, ICU patients.

Variables	SBEN Group (*n* = 48)	NGEN Group (*n* = 65)	*p*-Value
Actual feeding volume (mL/day)	992.88 (706.44–1191.38)	824.50 (541.38–1054.88) *	0.025 *
Actual energy intake (kcal/day)	921.15 (635.79–1172.81)	707.40 (487.24–949.39) **	0.004 **
Actual protein intake (g/day)	36.37 (25.43–46.91)	28.30 (19.49–37.98) **	0.005 **
Actual protein intake (g/kg/BW)	0.60 (0.41–0.74)	0.49 (0.31–0.69) *	0.029 *
Energy intake achievement rate (%)	60.29 (41.29–75.11)	49.02 (32.36–66.74) *	0.026 *
Protein intake achievement rate (%)	49.63 (34.16–62.01)	40.85 (25.78–57.18) *	0.029 *

Values are median (interquartile range (IQR)). Mann–Whitney U test. * *p* < 0.05, ** *p* < 0.01.

**Table 3 nutrients-12-02009-t003:** Comparison of survivors and nonsurvivors according to hospital mortality in malnourished, critically ill, ICU patients.

Variables	Nonsurvival (*n* = 49)	Survival (*n* = 64)	*p*-Value
Group (*n*, %) ^a^			1.000
SBEN group	37 (75.51%)	27 (42.19%)	
NGEN group	22.59 (57.14%)	37 (57.81%)	
Age (year)	72.13 (55.47–83.26)	73.03 (60.54–82.22)	0.498
Gender—Male (*n*, %) ^a^	37 (75.51%)	42 (65.63%)	0.353
Body mass index (kg/m^2^)	22.59 (20.44–26.67)	23.52 (21.41–26.21)	0.412
mNUTRIC score	7.00 (6–8)	7.00 (5–8)	0.255
APACHE II score	30.00 (26.5–34)	28.00 (24–32)	0.176
SOFA—day 1	10.00 (8–13.5)	10.00 (6–12.75)	0.257
SOFA—day 3	11.00 (8–13)	9.00 (6–12)	0.082
SOFA—day 7	10.00 (7–13)	8.00 (6–10) **	0.007 **
CCI	1.00 (0–3)	2.00 (0–3.75)	0.333
Comorbidity (*n*,%) ^a^			
Diabetes mellitus	11 (22.45%)	22 (34.38%)	0.241
Chronic kidney disease	13 (26.53%)	10 (15.63%)	0.234
Cardiovascular disease	20 (40.82%)	28 (43.75%)	0.904
Cancer	12 (24.49%)	18 (28.13%)	0.827
Liver cirrhosis	19 (38.78%)	28 (43.75%)	0.735
Obstructive pulmonary disease	17 (34.69%)	20 (31.25%)	0.854
Average of energy intake achievement rate (%) post-enrollment			
2nd day	43.69 (24.9–58.88)	49.44 (35.44–70.69) *	0.042 *
3rd day	40.63 (24.43–63.04)	55.91 (38.98–72.22) **	0.009 **
4th day	41.45 (27.78–66.7)	59.76 (40.3–76.73) **	0.005 **
5th day	44.55 (26.86–68.16)	60.80 (44.13–84.31) **	0.002 **
6th day	45.97 (24.78–66.57)	63.45 (48.93–83.84) **	0.001 **
7th day	48.99 (25.01–67.34)	66.07 (50.68–84.47) **	<0.001 **

Values are median (IQR). Mann–Whitney U test. ^a^ Chi-squared test. * *p* < 0.05, ** *p* < 0.01.

**Table 4 nutrients-12-02009-t004:** Cox regression analysis for factors associated with hospital mortality.

Variables	Univariate Analysis	Multivariate Analysis
	HR (95% CI)	HR (95% CI)
Age (year)	0.99 (0.98–1.01)	
Group (SBEN vs. NGEN)	0.56 (0.31–0.997) *	0.65 (0.36–1.18)
Gender (Female vs. Male)	0.62 (0.32–1.20)	
Body mass index (kg/m2)	0.97 (0.91–1.04)	
mNUTRIC score	0.91 (0.72–1.14)	
APACHE II score	1.01 (0.96–1.06)	
SOFA—day 1	0.99 (0.92–1.07)	
SOFA—day 3	1.05 (0.97–1.13)	
SOFA—day 7	1.12 (1.03–1.22) **	
Energy intake achievement rate (%) (3 days after the index day) (<65% vs. ≥65%)	2.53 (1.25–5.11) *	2.29 (1.12–4.69) *

Cox regression. * *p* <0.05, ** *p* < 0.01.

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
