# Peer review of "Comparison of Feeding Efficiency and Hospital Mortality between Small Bowel and Nasogastric Tube Feeding in Critically Ill Patients at High Nutritional Risk"

_nutrients, 2020, doi:10.3390/nu12072009_

Round 1

Reviewer 1 Report

It is a very well presented study so there are few criticisms to make.
It is a retrospective study, so its conclusions should be taken with caution, but this problem cannot be corrected.
I would just need you to clarify the criteria by which patients received enteral nutrition SBEN (N = 48) or NGEN (N = 65). I don't know if you compare the current SBEN results with the NGEN from previous years. Or if there are some doctors who always use SBEN and others NGEN.

Reviewer 2 Report

This present study compared the feeding efficiency and hospital mortality of two feeding routes (maintaining NGEN or shifting to SBEN) on the seventh ICU day in respiratory ICU patients who are undernourished and at high nutritional risk.

All feeding parameters were signify higher in SBEN group than NGEN group. Multivariate analysis indicated that Achievement rate < 65 % of the energy requirements on day 3 was the only factor independently associate with mortality.

  1. Authors performed this study to clarify whether SBEN improved the mortality compared to NGEN. They should state the reason why there was no significant difference in the mortality between SBEN group and NGEN group in spite of the higher rate of energy intake achievement in SBEN group. In Page 9, they stated that this study included patients with higher SOFA score. However, there was no significant difference in SOFA score between two groups. 
  2. Authors should describe the exact reason why the rate of energy intake achievement was lower in NGEN group than SBEN group. It is better to show the actual number of gastroesophageal reflux and aspiration pneumonia in both groups, if possible.
  3. It is better to show the reason why doctors retained NGEN instead of shifting to SBEN in treating undernourished patients, if possible.

Reviewer 3 Report

You have already published the abstract of this manuscript with similar but not identical authorship and wording. You should explain this in the manuscript.

https://academic.oup.com/cdn/article/4/Supplement_2/1152/5844504

In table 1 legends should be added to explain what do numbers represent numeric variables and categorical variables.

Why didn't you present table 3 stratified also by study groups? because univariate cox showed a difference in survival.

Legend of table 3 is incomplete as for statistical tests employed.

In table 4 is not explained what do numbers represent.
